# A 3D Bio-Printed-Based Model for Pancreatic Ductal Adenocarcinoma

**DOI:** 10.3390/diseases12090206

**Published:** 2024-09-10

**Authors:** Claire Godier, Zakaria Baka, Laureline Lamy, Varvara Gribova, Philippe Marchal, Philippe Lavalle, Eric Gaffet, Lina Bezdetnaya, Halima Alem

**Affiliations:** 1IJL, CNRS, Université de Lorraine, 54000 Nancy, France; claire.godier@univ-lorraine.fr (C.G.); zakaria.baka@univ-lorraine.fr (Z.B.); eric.gaffet@univ-lorraine.fr (E.G.); 2CRAN, CNRS, Université de Lorraine, 54506 Vandœuvre-lès-Nancy, France; la.lamy@nancy.unicancer.fr (L.L.); l.bolotine@nancy.unicancer.fr (L.B.); 3Département Recherche, Institut de Cancérologie de Lorraine (ICL), 6 Avenue de Bourgogne, 54519 Vandœuvre-lès-Nancy, France; 4Institut National de la Santé et de la Recherche Médicale (INSERM) U1121, Biomaterials and Bioengineering, 1 rue Eugène Boeckel, 67100 Strasbourg, France; varvara.gribova@gmail.com (V.G.); philippe.lavalle@inserm.fr (P.L.); 5Faculté de Chirurgie Dentaire, Université de Strasbourg, 8 rue Sainte Elisabeth, 67000 Strasbourg, France; 6LRGP, CNRS, Université de Lorraine, 54000 Nancy, France; philippe.marchal@univ-lorraine.fr; 7Institut Universitaire de France, 75000 Paris, France

**Keywords:** 3D bio-printing, pancreatic ductal adenocarcinoma, co-culture

## Abstract

**Simple Summary:**

Pancreatic ductal adenocarcinoma (PDAC) has a very poor prognosis, partly because existing preclinical models do not accurately recreate the tumor microenvironment. This study explores the use of 3D bio-printing, a cutting-edge technology, to develop more realistic cancer models. By combining pancreatic cancer cells and cancer-associated fibroblasts in a hydrogel, we have focused our efforts on constructing a PDAC, resulting in viable, proliferating tumors with heterogeneous composition. The findings suggest that 3D bio-printing can produce tumor models that maintain cell viability and offer a versatile platform for improving cancer therapy research.

**Abstract:**

Pancreatic ductal adenocarcinoma (PDAC) is a disease with a very poor prognosis, characterized by incidence rates very close to death rates. Despite the efforts of the scientific community, preclinical models that faithfully recreate the PDAC tumor microenvironment remain limited. Currently, the use of 3D bio-printing is an emerging and promising method for the development of cancer tumor models with reproducible heterogeneity and a precisely controlled structure. This study presents the development of a model using the extrusion 3D bio-printing technique. Initially, a model combining pancreatic cancer cells (Panc-1) and cancer-associated fibroblasts (CAFs) encapsulated in a sodium alginate and gelatin-based hydrogel to mimic the metastatic stage of PDAC was developed and comprehensively characterized. Subsequently, efforts were made to vascularize this model. This study demonstrates that the resulting tumors can maintain viability and proliferate, with cells self-organizing into aggregates with a heterogeneous composition. The utilization of 3D bio-printing in creating this tumor model opens avenues for reproducing tumor complexity in the future, offering a versatile platform for improving anti-cancer therapy models.

## 1. Introduction

Pancreatic cancer is designated as the fourth leading cause of cancer mortality in Western countries [1,2,3]. The rapid tumor invasion by lymphatic and nerve pathways, the poor prognosis related to late diagnosis, and the lack of early and reliable biological markers detection are major challenges that the scientific community has to tackle. Moreover, exocrine pancreatic cancer represents 90% of malignant pancreatic tumors (ductal adenocarcinoma—PDAC) [4]. In the context of PDAC, the establishment of metastatic dissemination is achieved, in particular, by an invasion of the lymphatic and nervous pathways conferred by the epithelial–mesenchymal phenotype of pancreatic cancer cells. A rapid epithelial–mesenchymal transition (EMT) allows the epithelial cells to lose their adherent characteristics and gain migratory and invasive properties, facilitating metastatic dissemination [5,6,7]. This phenomenon is more accentuated by the presence of a dense stroma rich in fibroblasts, which play a crucial role in tumor progression and drug resistance. Regarding vascularization in PDAC, this dense stroma can hinder blood vessel distribution and create areas of low perfusion, worsening hypoxia and leading to hypovascularization, resulting in drug resistance [8].

The only treatments currently available are excisional surgery (affecting only 15% of patients) and chemotherapy, with the expected survival rate of patients following these treatments being very low (less than 10% 5-year survival) [2,9]. Despite the scientific community’s efforts to develop new treatments, the failure rate of new anti-cancer molecules remains above 90% [10], with development taking 10 to 15 years [11]. This represents a significant financial investment and underscores the need for new reliable and predictive preclinical models [12].

Currently, the development of new drugs is mainly based on the use of two-dimensional (2D) cell culture and animal models. It is obvious that even if the 2D cell culture models have been proven for years, they only partially reflect the tumor complexity, mostly due to its monolayer cell organization [9]. In addition, in vivo animal testing also raises translational concerns. Indeed, the metabolic pathways involved in animals are not the same as those present in humans, which can create differences when molecules are transferred to the clinical phase [13,14]. The three-dimensional (3D) models have been shown to overcome these limitations through the development of organoids, spheroids, human xenografts, and explants [15,16,17]. Lazzari et al. (2018) [18] attempted to integrate human umbilical vein endothelial cells (HUVECs) into PDAC spheroids but failed to form blood vessel-like structures. Kuen et al. (2017) [19] included macrophages in PDAC spheroids, showing their infiltration and adoption of a tumor phenotype. Liu et al. (2021) [20] developed a heterospecific spheroid model to study the interactions between pancreatic tumor cells and stellate cells, simulating the characteristics of CAFs. Other studies, such as those by Lee et al. (2020) [21], used organoid models to reproduce the PDAC tumor microenvironment and observed an increase in cell migration and expression of tumor markers in the presence of stroma. Nevertheless, their engineering still faces challenges in terms of poor reproducibility in size and architecture [22]. Furthermore, there exist in vitro three-dimensional (3D) models derived from patient cells, including patient-derived organoids (PDOs) and patient-derived explants (PDEs). PDOs are three-dimensional structures cultivated from tumor cells isolated from patients, while PDEs consist of segments of whole tumor tissue that are directly obtained from patients and subsequently cultured in an in vitro environment. PDOs, for example, offer a closer approximation of the patient’s tumor with respect to genetic and phenotypic characteristics, rendering them highly valuable for personalized medicine approaches. Conversely, PDEs preserve the tissue architecture and microenvironment, facilitating the investigation of tumor–stroma interactions within a more natural context. Both PDOs and PDEs surpass traditional cell line models by maintaining the heterogeneity of the tumor, which is essential for comprehending treatment responses and resistance mechanisms. Additionally, these models can be employed for high-throughput drug screening, yielding more reliable and predictive data for preclinical assessments; however, they necessitate prolonged cell culture periods and present challenges in accessibility compared to immortalized cell lines. The effort to constantly improve cancer models has turned to the 3D bio-printing technique in the last decade [13,23,24]. This technique allows for overcoming the issues encountered in the development of classical 3D models by providing complete control over the size, architecture, and composition of the synthesized tumor mass [25,26]. Extrusion-based 3D bio-printing is mostly used as it enables (i) the deposition of multiple or selected cell types and (ii) working in compatible laboratory workflows, thus building multimaterial multicellular 3D models that reproduce physiological features of the tumor. The potential of extrusion bio-printing to recreate the tumor microenvironment within in vitro models has been recently reviewed by Flores-Torres et al. (2023) [27]. The focus of 3D bio-printing has been almost exclusively on breast, ovarian, and brain cancers [28,29,30,31,32,33,34]. Only a limited number of 3D bio-printing models of PDAC have been reported so far. A laser-assisted bio-printing model comprising different lineages of pancreatic cells turned out to be an attractive 3D cell model for studying the tumorigenesis of PDAC [9]; another report by Langer et al. (2019) [35] addressed extrusion-based bio-printing using patient-derived xenograft-derived PDAC. The authors demonstrated that patient-derived PDAC bio-printed tissues can recapitulate aspects of neoplastic tissues in vivo. The recent study by Sgarminato et al. (2023) [36] proposed an original approach to creating a 3D model of PDAC using melt electrowriting and conducted a layer-by-layer deposition of different cell types without mixing them together before the deposit. Many more studies performed by 3D bio-printing were reported in the context of dysfunction of the endocrine part of the pancreas, causing diabetes [37,38]. One of the best models describing the generation of the PDAC model by bio-printing is the one built by Hakobyan et al. (2020) [9], and it is restricted by its monoculture with only pancreatic cells, creating a lack of interaction with other cells of the microenvironment. To the best of our knowledge, only one work describes the bio-printing by extrusion of pancreatic and fibroblasts for PDAC. This work describes a process developed to generate PDAC models by bio-printing cell-laden gelatin methacryloyl (GelMA) hydrogel beads through dot extrusion printing [39]. The cells used in this work were the human pancreatic cancer cell line (BxPC-3) and normal human dermal fibroblast cells (NHDFs) to better mimic the tumor microenvironment. Another alternative is to use direct co-culturing of tumor cells with cancer-associated fibroblasts (CAFs). Such an approach can shorten the fabrication time of relevant tumor models, address different tumor stages, and overall provide a high-throughput platform for anti-cancer drug screening. CAF-enriched models indeed have many similarities with tumors in terms of up-regulated genes, cytokine production, growth factors release, etc. [35,36,37]. A 3D bio-printed model of ovarian cancer cells and myofibroblasts, MeWo (the latter stands for CAFs), was also recently described and characterized [31]. The interesting work of Wei et al. (2024) [40] also presents a 3D extruded bio-printed model using a GelMA suspension bath where cell-laden collagen/gelatin beads were directly deposited to generate spherical multicellular aggregates, leading to dense spheroids composed of Panc-1 cells and fibroblasts.

In the present study, we reported the development of an innovative preclinical 3D model using the 3D extrusion bio-printing technique to combine several cell types, pancreatic cancer cells (Panc-1) and cancer-associated fibroblasts (CAFs) encapsulated in a sodium alginate and gelatin hydrogel printed at 37 °C, to best mimic the PDAC tumoral microenvironment. The combination of gelatin/alginate-based bio-printing was preferred as it offers an attractive alternative to GelMA due to its biocompatibility (no UV cross-linking is necessary), ability to retain cells, flexibility in ink formulation, and potentially lower cost. These advantages make it a promising option for further manufacturing complex tissues in the context of cancer therapy. Our objective with this experimental model was to simulate the pancreatic ductal adenocarcinoma using the Panc-1 cell line enriched with CAFs to closely study the remarkable epithelial–mesenchymal plasticity of Panc-1 cells. This model was characterized by several viability tests, such as Live/Dead staining, the WST-1 test, and the AlamarBlue test, to demonstrate the presence of metabolic activity. Histological analysis revealed a good homogeneity of the cellular distribution within the developed structure. Immunological characterization also highlights the proliferative capacity and accurate identification of each cell type. The possibility of further enriching the proposed model by integrating endothelial cells to form a vascular network would enhance the complexity of the tumor model. Although the preliminary results are promising, this approach still requires in-depth investigations to evaluate its impact on tumor behavior and response to treatments. All these elements make the proposed 3D model a relevant, robust, and promising model, offering the scientific community a new preclinical model.

## 2. Materials and Methods

### 2.1. Reagents and Cell Lines

The Panc-1 (pancreatic ductal adenocarcinoma) cell line and MeWo cells (granular fibroblasts derived from human melanoma) used as cancer-associated fibroblasts (CAFs) were purchased from ATCC (Cat. No. CRL-1469 for Panc-1 and Cat. No. ATCC HTB65™ for MeWo cells). TTFLUOR HUVEC (Green Fluorescent Human Umbilical Vein Endothelial Cells) were supplied by Innoprot. Cell culture media and reagents, including Dulbecco’s Modified Eagle Medium (DMEM), Minimum Essential Medium (MEM), Endothelial Cell Growth Medium (ECGM), fetal bovine serum (FBS, Cat. No. F7524), Trypsin-EDTA solution, penicillin–streptomycin, amphotericin B, L-glutamine, Dulbecco’s Phosphate-Buffered Saline (DPBS), Hank’s Balanced Salt Solution (HBSS), sodium pyruvate, paraformaldehyde 4%, sodium alginate, gelatin (type B, bovine origin), calcium chloride, and sodium citrate were all supplied by Sigma-Aldrich. The Live/Dead Viability/Cytotoxicity Kit for mammalian cells was purchased from ThermoFisher Scientific. Cell proliferation kits, including the water-soluble tetrazolium 1 (WST-1) kit and the AlamarBlue kit, were purchased from Roche Laboratories.

### 2.2. Rheological Characterization

Rheological measurements were performed using a rotational rheometer (ARES-G2, TA instruments-Waters, USA, Equipped with oven and Nitrogen) with parallel plate geometry (40 mm flat plate). All measurements were recorded with a 1.0 mm gap width at 37 °C. The viscosities and shear stress of the gelatin–alginate hydrogel were assessed with a flow curve in the range of 1.0 to a 100 s^−1^ shear strain rate. The dynamic sweep test was performed at a shear strain of 1.0 to 100 s^−1^ to determine the solid- and liquid-like state of the hydrogel.

### 2.3. Cell Culture

DMEM supplemented with 10% FBS, 2% L-glutamine, 1% penicillin–streptomycin, and 0.05% amphotericin B was used to grow Panc-1 cells. For MeWo cells, MEM supplemented with 10% FBS, 2% L-glutamine, 1% penicillin–streptomycin, 0.05% amphotericin B, and 1 mM sodium pyruvate were used. The culture medium for HUVEC cells consisted of an all-in-one ready-to-use ECGM medium supplemented with 1% penicillin–streptomycin and 0.05% amphotericin B. T75 flasks were used for cell growth at 37 °C in an atmosphere of 5% CO_2_ and were passaged twice per week to enable exponential cell growth.

### 2.4. Tumoral Mass Design

The tumor model was designed as a cylinder with a diameter of 6 mm and a thickness of 1.5 mm using the One Shape online 3D computer-aided design (CAD) platform.

### 2.5. Bio-Ink Preparation and Tumoral Mass 3D Bio-Printing

The term “bio-ink” refers to the hydrogel in which the cells are incorporated. To prepare the bio-inks, gelatin and alginate powder were sterilized for 60 min under UV (254 nm) and then diluted with the corresponding cell medium to achieve a final concentration of 15/2% gelatin/sodium alginate. The final solutions were maintained under sterile conditions with magnetic stirring at 37 °C overnight. The following day, the cells were trypsinized, suspended in an appropriate culture medium, and integrated into the hydrogels. The resulting bio-inks were gently agitated to ensure a uniform distribution of cells. The cell compositions of the various bio-inks chosen for this study are presented in Table 1.

For the 3D bio-printing process, the bio-inks were carefully loaded into 3 mL cartridges, kept at room temperature for 15 to 20 min, and then placed into the print head of the bio-printer. The bio-printer used is the Bio X with three different printer heads supplied by Cellink (https://www.cellink.com/bioprinting/bio-x-3d-bioprinter). This procedure (refer to Figure 1) was sufficient to prepare approximately 48 samples (2x 24-well plates) with high reproducibility in terms of size and form in less than 4 h of handling. The resulting tumor model from this process is a cylinder of 6 mm diameter and 1.5 mm thickness, with a volume of 62.5 µL of bio-ink for each structure. This is a quick approach that can be useful for high-throughput screening applications. The bio-printing parameters are defined in Table 2.

The bio-printing process involved trypsinizing cells and embedding them in hydrogel (2% sodium alginate + 15% gelatin) to create bio-ink. Tumor models were printed as cylinders measuring 6 mm in diameter and 1.5 mm thick, with a volume of 62.5 µL of bio-ink for each structure. The printed structures were then cross-linked (using CaCl_2_ 100 mM for 7 min), supplemented with fresh culture medium, and placed in an incubator at 37 °C with 5% CO_2_ until the experiments were conducted.

### 2.6. Cellular Viability

The cell viability assay was conducted following the manufacturer’s instructions for the Viability/Cytotoxicity Kit for Live/Dead Mammalian Cells. The 3D bio-printed structures were rinsed with DPBS and then treated with a solution containing 500 µL of 4 µM calcein and 16 µM homodimer (EthD1) after removing the culture medium completely. Subsequently, the structures were incubated in the dark at 37 °C for 40 min and examined using a confocal microscope (Zeiss LSM 710, Heidelberg, Germany), as depicted in Table 3.

The experiments were performed on days 1, 3, and 7 after bio-printing. For each time point, negative controls (NCs) (structures containing only dead cells) were prepared by exposing the bio-printed structures to 70% methanol for 30 min before treatment with Live/Dead kit reagents.

### 2.7. Metabolic Activity and Cell Proliferation

To further quantify the cellular metabolic activity in bio-printed tumor models and its evolution over time, WST-1 and AlamarBlue tests were conducted. After removing the culture medium, the bio-printed structures were supplemented with a fresh culture medium containing 10% (*v*/*v*) WST-1 reagent or AlamarBlue reagent and incubated at 37 °C for 5 h. Subsequently, 500 µL of 1.5% sodium citrate solution was added to each structure for 30 min to dissolve the matrix and release the dye produced by the cells. The resulting solution was transferred into 96-well plates (100 µL/well), and optical densities at wavelengths of 450 nm and 630 nm for the WST-1 test and fluorescence at wavelengths of 544 nm and 590 nm for the AlamarBlue test, were determined using a microplate reader. The experiments were conducted on days 1, 3, and 7 after bio-printing.

### 2.8. Histological and Immunohistochemical Analysis

For histological analysis, bio-printed tumor models were processed according to the protocols provided by Cellink.

The samples were first washed with HBSS (Hank’s Balanced Salt Solution) for 15 min at 37 °C and then fixed in a 4% paraformaldehyde solution containing 50 mM CaCl_2_ for 1 h at room temperature.

After fixing, the structures were washed twice with the HBSS solution for 5 min before being dehydrated successively in 70% ethanol, 96% ethanol, 100% ethanol, and 100% xylene baths.

The samples were then embedded in paraffin, cut into 6 µm thick sections using a microtome, and stained with hematoxylin–eosin–saffron (HES) using the DAKO CoverStainer^®^ [41].

Structures were examined on days 1, 3, and 7 after bio-printing. At least three independent experiments were conducted, and a minimum of three different images were analyzed per sample at each time point. For immunohistological studies, the bio-printed structures were fixed and embedded in paraffin following the protocols provided by Cellink. Subsequently, the samples were sectioned into 6 µm thickness using a microtome, and immunological staining was carried out using the DAKO Omnis^®^ IHC automate.

The following markers were used: Ki67 [42] for cell proliferation, cytokeratin 19 (CK19) [43] for pancreatic cell staining, HMG-box 10 (SOX10) for MeWo cells [44], and vimentin for mesenchymal cell types [45,46]. The specific antibodies used are listed in Table 4.

The establishment of a vascular network within the bio-printed structures (composed of Panc-1/MeWo/TTFLUOR HUVEC) was performed using the fixation protocol for cryosections provided by Cellink.

-The structures were first washed twice with Hank’s Balanced Salt Solution (HBSS) for 10 min at 37 °C and then fixed in a 4% paraformaldehyde solution containing 50 mM CaCl_2_ for 2 h at room temperature at each time point.-After fixation, the structures were washed twice with HBSS for 10 min each and then placed in a fresh HBSS solution at 4 °C for 45 min.-Afterwards, the structures were exposed to a treatment with 30% sucrose for 45 min at room temperature. The bio-printed structures were embedded in ShandonTM CryomatrixTM resin, frozen at −80 °C, and then sectioned.-The samples were cut into 10 µm thick sections using a cryostat with an enclosure set at −20 °C.

The structures were observed on days 1, 3, and 7 after bio-printing under an epifluorescence microscope (Olympus AX-70) using appropriate filters, as depicted in Table 5.

## 3. Results

### 3.1. Rheological Characterization

To ensure the suitability of the proposed ink for 3D bio-printing applications, the rheological behavior of the hydrogel solution was studied before 3D bio-printing at 37 °C, the temperature at which the entire bio-printing process was conducted. Figure 2 presents the viscosity and shear stress evolution depending on the shear rate. The yield stress and shear-thinning properties of the alginate–gelatin bio-ink could be determined. At low shear rates, between 0.01 and 0.1 s^−1^, the shear rate stress remains constant, indicating solid-like behavior with a static yield strength of 100 Pa. This rheological behavior is crucial during 3D bio-printing by extrusion to preserve the shape of the bio-printed structure during and after material deposition. Beyond this low shear rate, the applied shear stress is sufficient to disrupt physical networks (resulting from the combination of supramolecular interactions within and between macromolecules). As the 3D bio-printing process was carried out at around 20 kPa at 37 °C, the bio-ink behaves like a liquid and can be easily extruded under these conditions. Finally, the decrease in viscosity with increasing shear stress confirms the shear-thinning behavior of the ink, which aligns well with Łabowska et al. (2021) [47].

### 3.2. Optimization of Co-Culture (Ratios and Concentration)

Cancer-associated fibroblasts (CAFs) account for around 80% of PDAC; therefore, our starting point was the optimization of the co-culture Panc1/MeWo model in terms of cell concentrations, keeping the constant ratio Panc1/MeWo 1:4 within the 3D matrix. Concerning the tri-culture model, it is acknowledged that PDAC tumors are poorly vascularized, therefore requiring a proportion of endothelial cells higher or equal to that of pancreatic cancer cells in order to form a vessel [48]. According to this information, Panc-1/MeWo/HUVEC ratios were fixed at 1:4:4.

A recent study demonstrated that the final cell concentration of 1·10^6^ cells·mL^−^^1^ was not sufficient to form a 3D cellularized object [30]. A cell concentration of 2·10^6^ cells·mL^−^^1^ also provided hardly visible cell density (Figure 3). When cell concentration was doubled to 4·10^6^ cells·mL^−^^1^, we could observe a 3D cellularized object formation; however, cell proliferation was too high and led to massive cell death over the days. We observed that the cells are more numerous and closer together, no longer limited by their interactions. However, after 7 days of culture following bio-printing, the majority of cells were dead. This mortality is visually assessed by the degradation of the nucleus, which appears to “burst”, and the rapid degradation of the extracellular matrix, visually assessed by the formation of pores represented by whitish areas on the periphery of the cells. An intermediate cell concentration of 3·10^6^ cells·mL^−^^1^ appeared as a good compromise between the two above-mentioned ones.

These results are in good agreement with previous work on 3D bio-printing with other types of cancer cells [9,27,31,34]. In these studies, the authors used cell densities around 1·10^6^ cells·mL^−^^1^, which is consistent with our range of values. For our further 3D bio-printed structure, we fixed a cell concentration of 3·10^6^ cells·mL^−^^1^ to maintain a good balance between cell viability and 3D structure integrity.

### 3.3. Proliferation and Cell Viability

Cell viability, which is essential for achieving functional cellular outcomes, was evaluated in bio-printed co-culture tumor models using the Live/Dead assay and confocal microscopy imaging [9,49]. The corresponding results are presented in Figure 4. One day after bio-printing, the majority of the cell population displayed green fluorescence, indicating high cell viability in the bio-printed structures. Significantly, this high viability, sustained on days 3 and day 7 post-bio-printing, aligns well with the findings of Liu et al. (2019) [50], who studied the cell viability of endothelial cells in co-culture within a hydrogel post-bio-printing, and with the research by Ermis et al. (2023) [6], who examined the viability of pancreatic spheroids in a hydrogel through 3D bio-printing. In this study, we observed an increase in the intensity of green fluorescence on day 7 compared to day 1. This increase can be attributed to the growth of the cell population through cell proliferation within the bio-printed structures. Additionally, there may be cell migration towards the periphery of the structures, necessitating further investigation. These results suggest that, visually, the 3D bio-printing process has no significant impact on cell viability.

To delve deeper, cell viability over time was studied by assessing the metabolic activity of cells in tumor models using the WST-1 and the AlamarBlue tests [6,50] (Figure 5). For both tests, the results are expressed as a percentage of cellular viability after numeric treatment and normalization to the highest value (on day 3) defined as 100% cellular viability. On day 1 after bio-printing, the tumor structures exhibited elevated metabolic activity for both WST-1 and AlamarBlue assays, corresponding to more than 80% of cellular viability (89% for AlamarBlue and 86% for WST-1). By day 3, WST-1 analysis and the AlamarBlue assay showed significantly higher levels of absorbance and fluorescence (respectively) than those observed on day 1, and this was maintained through day 7 (from 100% on day 3 to approximately 90% on day 7 for the WST-1 trial and to approximately 80% on day 7 with the AlamarBlue assay), indicating that the cells survived the bio-printing process, maintaining their metabolic activity. This continuous metabolic activity is attributed to the increase in the number of cells, reflecting cell proliferation within the bio-printed structures. It is also important to highlight that from day 3 to day 7, viability tends to decrease by about 10%. The decrease can be explained by the fact that over time, cells will escape the matrix and proliferate, leading to a major death related to the increasing number of cells compared to day 1 after bio-printing. Results consistent with this observation have been documented in the study conducted by Chaji et al. (2020) [32], where they illustrated a decrease in metabolic activity between day 2 and day 10 after bio-printing. This trend coincides with our own finding, where a similar reduction in metabolic activity was already observed as early as the 7th day after bio-printing. By consolidating all our results, we can confirm that the 3D bio-printing process applied to the co-culture in the hydrogel only slightly affects the viability and the metabolic activity of the cells.

### 3.4. Histological Analysis

Our model underwent histological analysis using HES staining to evaluate the morphology and distribution of cells in the matrix and to identify features indicative of PDAC. Histological characterization is of utmost importance, emphasizing the invasive nature of PDAC, as highlighted by its histological aggressiveness according to Buscail et al. (2012) [1]. This observation is supported by a series of PDAC histological studies, including those by Feig et al. (2012) [51], Swayden et al. (2019) [52], Liu et al. (2021) [20], and Koltai et al. (2022) [53], which demonstrated the abundance of stroma marked by an excess of extracellular matrix, known as desmoplasia. Jang et al. (2021) [7] noted that excessive stromal content is a significant histopathological feature of the disease. Lastly, the use of histological analyses to characterize a tumor model is common practice, as demonstrated in the works of Hakobyan et al. (2020) [9] and Novak et al. (2021) [41].

As shown in Figure 6, on day 1 after bio-printing, the cells were evenly distributed in the matrix, indicating homogeneous cell encapsulation in the prepared bio-inks. While maintaining their proliferative activity, we observed from day 3 to day 1 that the cells started forming increasingly massive clumps within the bio-printed structures. The cells spontaneously aggregated to form spheroid-like structures, maintaining cell–cell interactions in the bio-printed tumor models. Interestingly, areas of matrix retraction surrounding the newly formed cell aggregates were observed, suggesting matrix remodeling by the cells in the bio-printed structures. A study by Mollica et al. (2019) [54] illustrated similar histological sections to those resulting from our research, showing the creation of cellular structures within a hydrogel mimicking a breast tumor. Comparing single culture images also provided insights into the growth and aggregation patterns of two cell types, which could be attributed to their different proliferation rates. MeWo cells proliferate faster (doubling time = 18 h) than Panc-1 cells (doubling time = 36 h). However, it is important to note that Panc-1 cells have a more pronounced natural tendency than MeWo to form aggregates in both 2D and 3D environments. Over time, the aggregates became more numerous and larger, as depicted in Figure 7. On the first day after bio-printing, the mean diameter of small aggregates (SDis) was 22 ± 6 μm, and the mean diameter of large aggregates (LDis) was 30 ± 8 μm. By day 7 after bio-printing, the mean SDi had increased to 40 ± 6 μm, while the average LDi reached 53 ± 8 μm. This trend of cells self-organizing into spheroid-like clusters within bio-printed structures has been observed by various research groups using different hydrogels and cancer cells [55,56,57]. As suggested by other authors, this phenomenon underscores the significant role of the 3D environment in organizing cancer cells [56].

### 3.5. Immunological Characterization

For further characterization of our bio-printed structures, immunohistochemical studies were performed. Immunohistochemistry, as highlighted by Lin et al. (2015) [58] and emphasized by the research of Handra-Luca et al. (2011) [45], has become a valuable auxiliary method in the identification and classification of pancreatic neoplasms. These studies underscore the randomly observed co-expression of cytokeratin and vimentin, which are associated with the acquisition of mesenchymal histological features. Initially, we assessed the expression of Ki-67, a marker that identifies cells in the proliferation phase. In our structures, cells expressed Ki-67 from day 3 to day 7 after bio-printing (Figure 8), indicating sustained cell proliferation up to the 7th day post-bio-printing. This finding aligns with and elucidates the results of the WST-1 and AlamarBlue metabolic tests. Cell proliferation results in an increased cell count and overall metabolic activity in the bio-printed structures. The results demonstrate that the 3D bio-printing model can proliferate as 3D spheroids, as shown by Yakavets et al. (2020) [59], who evaluated a 3D spheroid model in co-culture (breast cancer cells + fibroblasts), illustrating proliferation on days 1, 3, and 7 after spheroid formation. Li et al. (2018) [60] and Langer et al. (2019) [35] also demonstrated this proliferation using the Ki-67 marker in spheroids embedded in hydrogel.

The expression of vimentin, a cytoskeletal protein typically expressed by cells of mesenchymal origin, including fibroblasts, was also assessed. MeWo cells and Panc-1 cells expressed this protein, with Panc-1 cells being vimentin positive in monoculture conditions (Figure 8). This result was expected, as Panc-1 cells are known to present an epithelial–mesenchymal phenotype, notably characterized by the expression of vimentin [44], and this specificity was reproduced in our models. The vimentin-positive staining supports our hypothesis that the epithelial–mesenchymal transition of Panc-1 cells can be promoted by the presence of CAFs.

In our bio-printed structures, we evaluated the expression of specific immunological markers CK19, specific to pancreatic cancer [38] and SOX10, a specific marker for MeWo cells from malignant melanoma [61]. We found strong CK19 expression in Panc-1 cells under monoculture conditions, while MeWo cells showed no expression (Figure 8). Similarly, in the co-culture structures, we identified cellular aggregates containing both CK19-positive cells (Panc-1 cells) and CK19-negative cells (MeWo cells).

MeWo cells exhibited strong positivity for SOX10 under monoculture conditions, while Panc-1 cells were negative (Figure 8). In the co-culture structures, we observed the coexistence of SOX10-positive cells and SOX10-negative cells, highlighting the heterogeneity of cell aggregates.

The observations unequivocally demonstrate that the bio-printed structures consist of heterotypic aggregates, wherein both cell types coexist. It is essential to highlight that the conclusions drawn at this juncture align with earlier discoveries by Baka et al. (2023) [31]. These findings affirm the viability of utilizing various cancer cell lines, distinct from SKOV-3, through their co-cultivation with MeWo.

### 3.6. Creation of a Vascular Network

The vascular network is known to play a critical role in the initiation and advancement of various types of cancers [12,62], including pancreatic cancer [63,64]. The existing literature indicates that typically, a vascular network begins to form around 7 days post-initiation [49,64]. In a study by Liu et al. (2019) [50], early vascular network formation was clearly observed during bone regeneration, utilizing the same endothelial cells (HUVEC in hydrogel) as in our study. However, it is noteworthy that pancreatic cancer is often associated with hypovascularization [8,65,66,67], which can impact vascular network development. Despite this, research has been directed towards investigating cell interactions in the context of pancreatic ductal adenocarcinoma (PDAC). A tri-culture model involving Panc-1, MeWo, and HUVEC cells was successfully maintained for 7 days, as indicated by green fluorescence visualization in cryosections under an epifluorescence microscope (Figure 9). Confocal microscopy was deemed unsuitable due to structural thickness and excessive background noise, leading to the adoption of cryosections as a more effective alternative. The successful evaluation of cell interactions within a tri-culture setting over a 7-day period highlights the harmonious coexistence of the three cell types within the bio-printed constructs. Further research endeavors are warranted to expand on these findings and gain deeper insights into the underlying mechanisms.

## 4. Conclusions

The innovative model outlined in this study enables the replication of the complexity and heterogeneity observed in cancerous tumors more accurately, thereby presenting novel opportunities for advancing research and development in anti-cancer therapies. The bio-printed model can be characterized using established two-dimensional methods. Detailed immunohistological investigations have allowed for the assessment of specific marker expressions such as Ki-67, CK19, SOX10, and vimentin to validate the tumor microenvironment relevance of the model. These analyses have shown that our bio-printed model displays significant cellular heterogeneity, effectively mirroring characteristics found in actual tumors. To our knowledge, this is the sole model that replicates the PDAC tumor microenvironment through 3D extrusion bio-printing, incorporating pancreatic cancer cells, cancer-associated fibroblasts (CAFs), and endothelial cells, all crucial components of the tumor microenvironment. A notable advantage of this bio-printed model is its adaptability to other cell types, including tri-cultures with immune cells that play a role in the immune response against anti-tumor treatments. Consequently, the proposed structure not only serves as a valuable tool for investigating the immunomodulatory response within the context of immunotherapeutic agents (whether cellular or material-based) but also as a robust platform for drug screening. This model is capable of assessing the efficacy of both established and novel therapeutic agents within a controlled environment that closely mirrors the complexity of the tumor microenvironment. Moreover, this model can be further enhanced by incorporating immune cells, increasing its biomimicry and allowing for more precise studies of tumor–immune system interactions, which are critical for advancing immunotherapy research. Our research team intends to utilize this model to evaluate the efficacy of gemcitabine, a well-known first-line treatment for PDAC, or to apply a Folfirinox chemotherapy regimen and compare its performance with other in vitro 3D models. Furthermore, the bio-printed model holds potential as a “chemogram”, facilitating personalized medicine by enabling the customization of treatment strategies based on the specific characteristics of an individual’s tumor. The versatility of this model makes it a powerful tool for studying various cancer types, offering critical insights into tumor behavior and treatment responses across different therapeutic modalities.

## Figures and Tables

**Figure 1 diseases-12-00206-f001:**
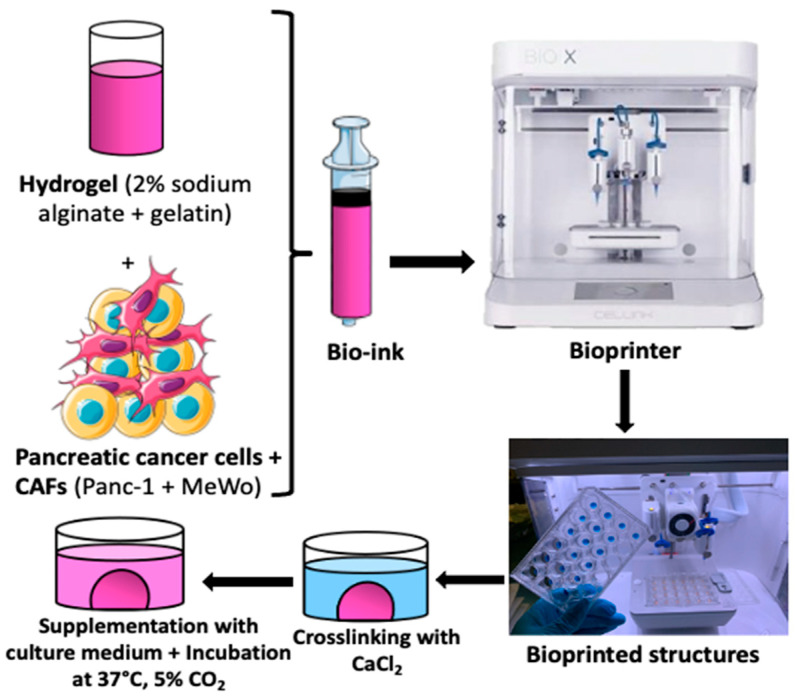
The bio-printing process involved trypsinizing cells and embedding them in hydrogel (2% sodium alginate + 15% gelatin) to create bio-ink. Tumor models were printed as cylinders measuring 6 mm in diameter and 1.5 mm thick, with a volume of 62.5 µL of bio-ink for each structure. The printed structures were then cross-linked (using CaCl_2_ 100 mM for 7 min), supplemented with fresh culture medium, and placed in an incubator at 37 °C with 5% CO_2_ until the experiments were conducted.

**Figure 2 diseases-12-00206-f002:**
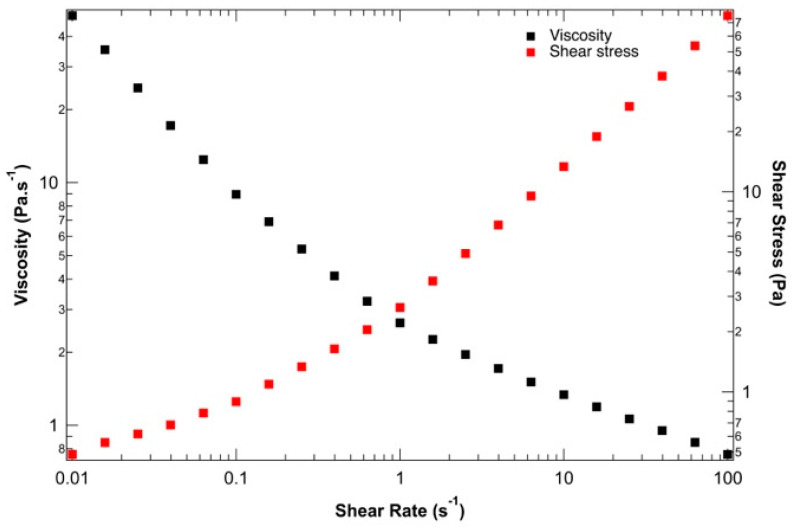
Evaluation of the shear-thinning behavior of bio-ink: viscosity (black) and shear stress (red) vs. shear strain rate.

**Figure 3 diseases-12-00206-f003:**
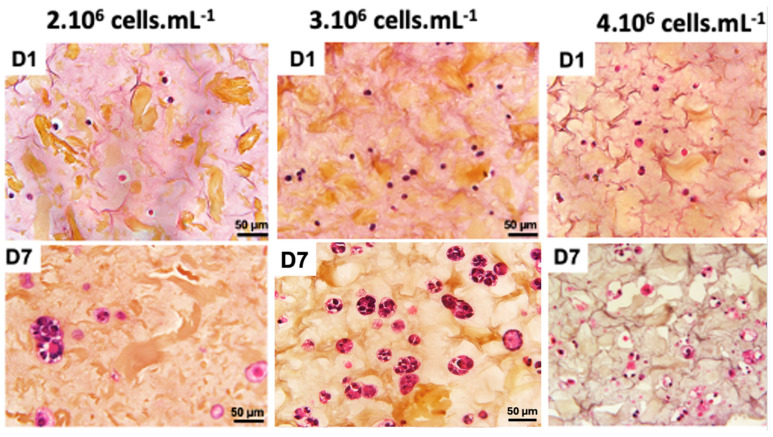
Optimization of co-culture cellular concentration Panc-1/MeWo cells (comparisons on days 1 and day 7 after bio-printing at concentrations of 2, 3, and 4·10^6^ cells·mL^−1^) by HES staining to evaluate the behavior of cells within the matrix.

**Figure 4 diseases-12-00206-f004:**
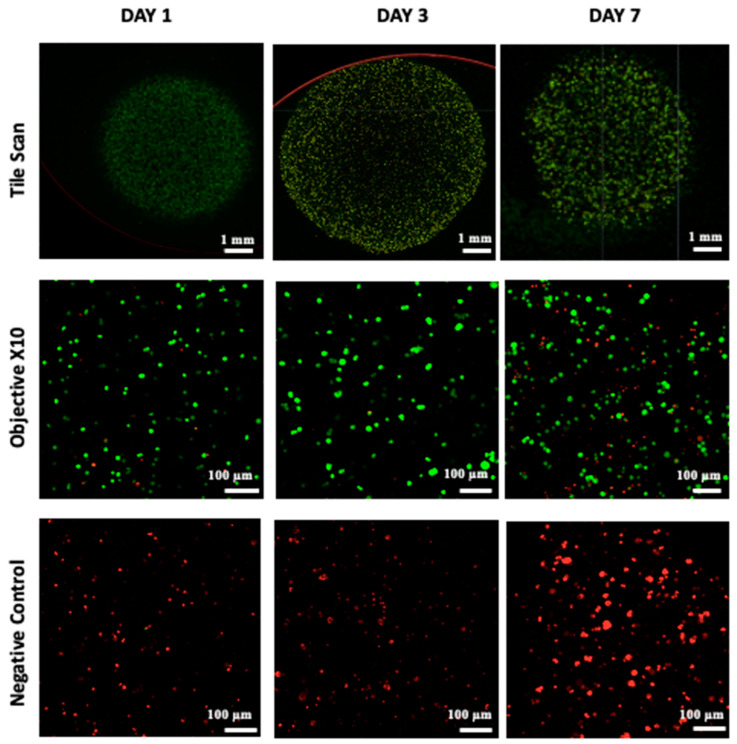
Live/Dead staining was conducted followed by confocal microscopy imaging on the bio-printed tumor models. Live cells emit green fluorescence, while dead cells emit red fluorescence. Tile scan images were obtained by combining images from multiple acquisition fields (10× objective) to provide an overall view of the bio-printed structure. NC designates negative control (bio-printed structures exposed to methanol for 30 min). Representative images from three independent experiments are displayed.

**Figure 5 diseases-12-00206-f005:**
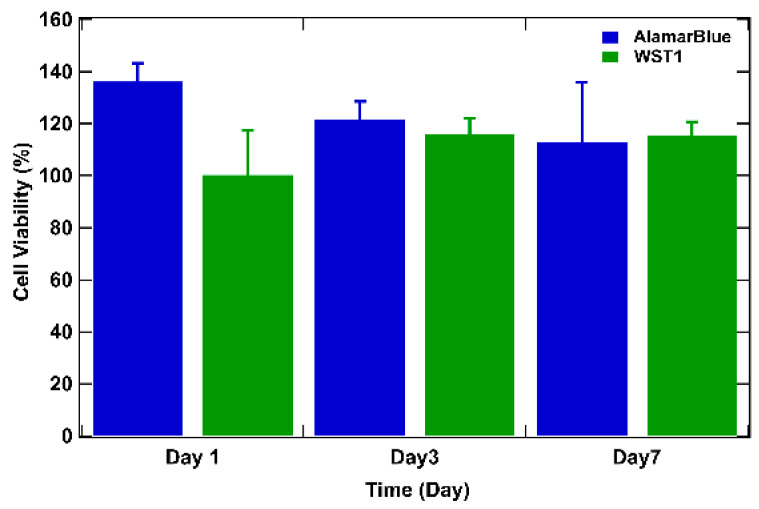
WST-1 and AlamarBlue assays performed on bio-printed structures to evaluate cellular activity over time. For both tests, the results are expressed as a percentage of cellular viability after numeric treatment and normalization to the highest value (on day 3), defined as 100% of cellular viability. Three independent experiments were performed. Errors bars correspond to standard deviations.

**Figure 6 diseases-12-00206-f006:**
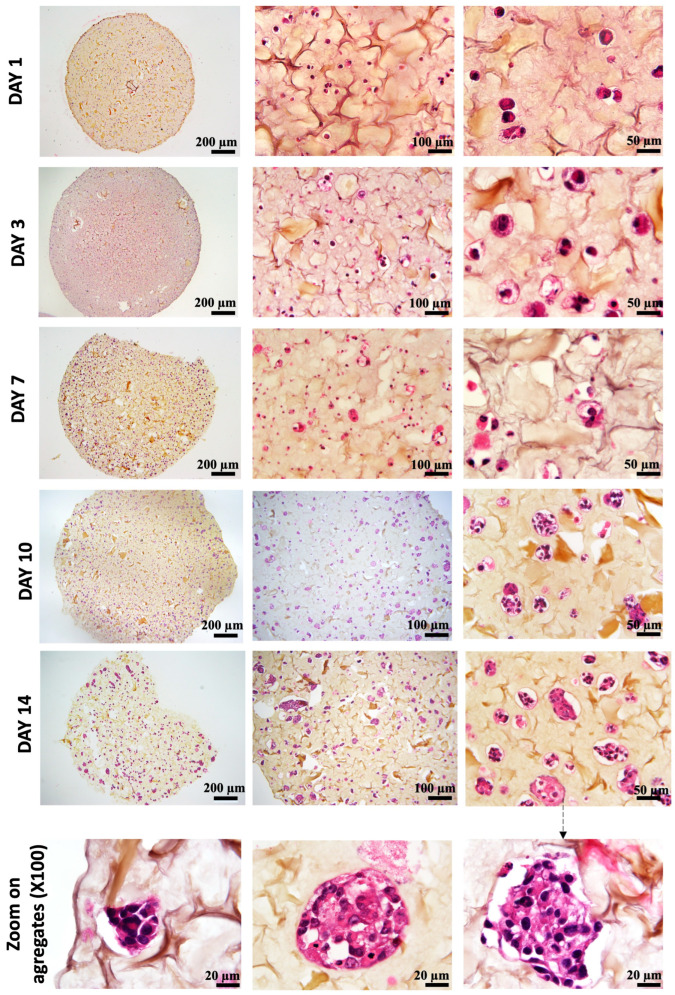
Histological analysis (HES staining) of bio-printed structures (co-culture condition) on days 1 to day 14 after bio-printing. Cells are individualized and homogenously distributed within the bio-printed structures. Purple stands for nuclear staining, while yellow/brown stains stand for collagen fibers. Representative images are shown.

**Figure 7 diseases-12-00206-f007:**
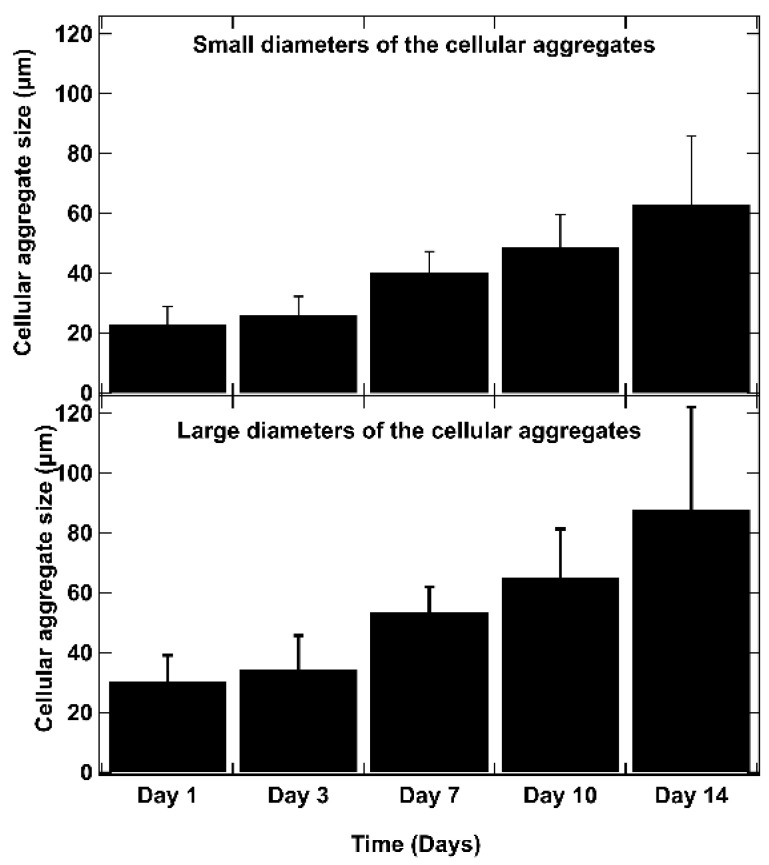
The evolution of aggregate dimensions over time in the bio-printed structures was analyzed. For each aggregate, a small diameter (SDi) and a large diameter (LDi) were defined. The small diameters (SDis) and large diameters (LDis) were determined using ImageJ image processing software. These measurements represent the diameters of the ellipse passing through the center of the ellipse, with the large axis (large diameter) (a) and small axis (small diameter) (b) intersecting at the central point of the ellipse. The SDi and LDi of 50 different aggregates are measured for each time point, and the average results are presented. Error bars indicate standard deviations. To confirm the statistical significance of the differences in cell diameters between small and large aggregates, we performed a Mann–Whitney U test at each time point. The calculated *p*-values are as follows: day 1: *p* = 0.0017, day 3: *p* = 0.0027, day 7: *p* = 0.0030, day 10: *p* = 0.0012, day 14: *p* = 0.0062. These results indicate statistically significant differences at each time point (*p* < 0.01).

**Figure 8 diseases-12-00206-f008:**
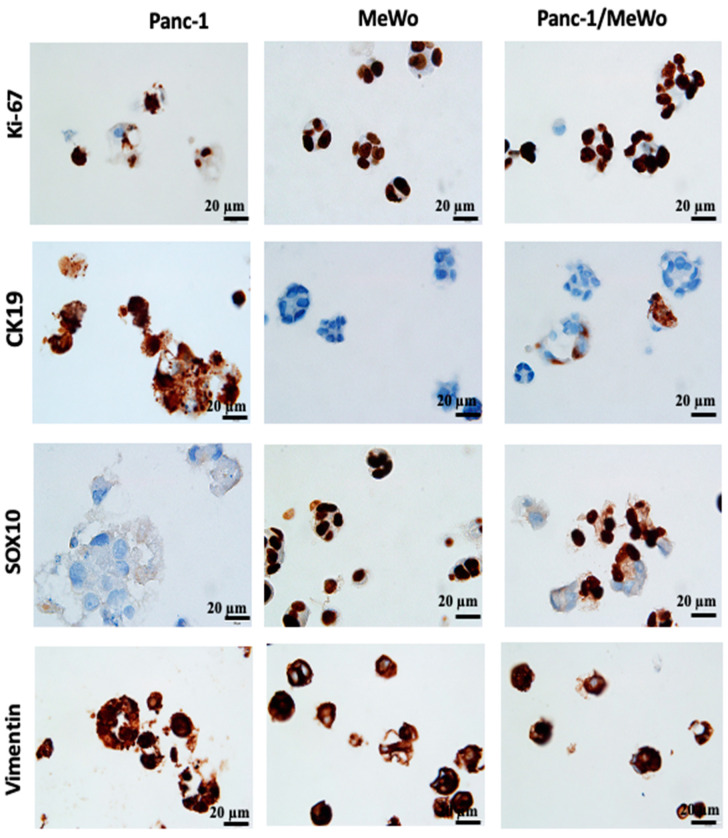
Immunohistochemistry (IHC) performed on 3D bio-printed structures to assess the expression of different markers. All sections were performed on day 7 after bio-printing. Both monoculture and co-culture conditions were analyzed. Representative images are shown.

**Figure 9 diseases-12-00206-f009:**
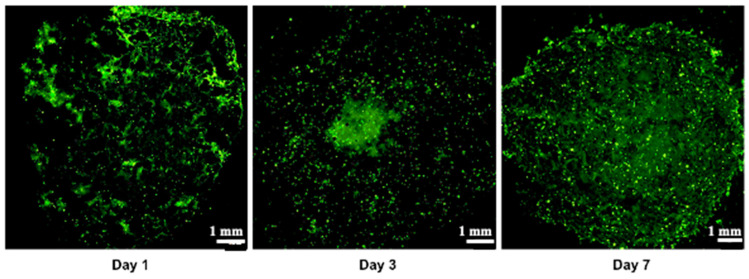
Vascularization assay demonstrating the tri-culture Panc-1/MeWo/HUVEC-GFP within the artificial matrix during 7 days of observation. These images represent 10 µm cryosections. The green color is associated with GFP-transfected HUVEC. Three independent experiments were performed. Representative images are shown.

**Table 1 diseases-12-00206-t001:** Composition of the bio-inks used in this study.

Bio-Ink Designation	Cell Type	Medium
P	Panc-1 cells	Complete DMEM (comprising 10% FBS)
M	MeWo cells	Complete MEM (comprising 10% FBS)
PM	Panc-1 cells + MeWo cells (1:4 ratio)	Complete DMEM + MEM (1:1 ratio)
PMH	Panc-1 cells + MeWo cells + HUVEC (1:4:4 ratio)	Complete ECGM

**Table 2 diseases-12-00206-t002:** Parameters for the 3D bio-printing process.

Parameters	Corresponding Value
Internal diameter of the printed needle	23 G (0.66 mm)
Printhead temperature	37 °C
Printing bed temperature	8 °C
Extrusion pressure	15–30 kPa
Printhead movement speed	5 mm·s^−1^

**Table 3 diseases-12-00206-t003:** Confocal microscope observation parameters for Live/Dead assays.

Reagent	Calcein	Ethidium Homo-Dimer 1
Excitation/emission wavelength (nm)	494/517 nm	528/617 nm
Standard set filter	Green channel: (EX/EM = 488/520 nm)	Red channel: (EX/EM = 561/596 nm)

**Table 4 diseases-12-00206-t004:** Antibodies used for IHC.

Antibody	Clone	Dilution	Positivity
SOX10	Monoclonal clone EP-268	1/200	MeWo
Anti-CK19	Monoclonal clone RCK108	1/100	Panc-1
Anti-Ki67	Monoclonal clone Mib1	1/50	MeWo Panc-1
Vimentin	Clone V9	1/200	Panc-1 MeWo

**Table 5 diseases-12-00206-t005:** Epifluorescence microscope parameters for vascularization tests.

Reagent	GFP
Excitation/emission wavelength (nm)	482/502 nm
Standard set filter	Green channel: (EX/EM = 460/510 nm)

## Data Availability

Suggested data availability statements are available from the corresponding author.

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
