# Peer review of "A 3D Bio-Printed-Based Model for Pancreatic Ductal Adenocarcinoma"

_diseases, 2024, doi:10.3390/diseases12090206_

Round 1

Reviewer 1 Report

Comments and Suggestions for Authors

Alem et al. propose an innovative approach for constructing a 3D model of PDAC and characterizing its corresponding features. This model is highly commendable as it incorporates crucial factors such as the tumor microenvironment. Currently, numerous in vitro 3D models have been extensively investigated, including PDO and PDE. However, the model presented by the author serves as a valuable complement to these existing models. Prior to acceptance of this article, I suggest implementing the following modifications.

1. The authors should emphasize the existing in vitro 3D models (such as PDO, PDE, ) in the introduction section and elucidate their comparative advantages over cell line models.

2. The following literature provides a good explanation of ‘The three-dimensional (3D) models have been shown to overcome these limits by the development of organoids and spheroids’ and may consider being cited.

A. Gu, J. Li, S. Qiu, S. Hao, Z.-Y. Yue, S. Zhai, M.-Y. Li, Y. Liu, Pancreatic cancer environment: From patient-derived models to single-cell omics. Mol. Omics 2024, 20, 220–233. DOI: 10.1039/D3MO00250K

3. Although the author does not provide application experiments, it is essential to include a discussion on potential application areas of this model and the author's future research directions at the conclusion of this article.

Comments on the Quality of English Language

minor revision

Author Response

Alem et al. propose an innovative approach for constructing a 3D model of PDAC and characterizing its

corresponding features. This model is highly commendable as it incorporates crucial factors such as

the tumor microenvironment. Currently, numerous in vitro 3D models have been extensively

investigated, including PDO and PDE. However, the model presented by the author serves as a valuable

complement to these existing models. Prior to acceptance of this article, I suggest implementing the

following modifications.

1.

The authors should emphasize the existing in vitro 3D models (such as PDO, PDE, ) in the

introduction section and elucidate their comparative advantages over cell line models.

Thank the reviewer for its valuable suggestion. We add a part concerning PDO and PDE with their comparative advantages over cell line models in the introduction (highlighted in yellow in the manuscript):

“Furthermore, there exist in vitro three-dimensional (3D) models derived from patient cells, including

patient-derived organoids (PDOs) and patient-derived explants (PDEs). PDOs are three-dimensional

structures cultivated from tumor cells isolated from patients, while PDEs consist of segments of whole

tumor tissue that are directly obtained from patients and subsequently cultured in an in vitro

environment. PDOs, for example, offer a closer approximation of the patient's tumor with respect to

genetic and phenotypic characteristics, rendering them highly valuable for personalized medicine

approaches. Conversely, PDEs preserve the tissue architecture and microenvironment, facilitating the

investigation of tumor-stroma interactions within a more natural context. Both PDOs and PDEs surpass

traditional cell line models by maintaining the heterogeneity of the tumor, which is essential for

comprehending treatment responses and resistance mechanisms. Additionally, these models can be

employed for high-throughput drug screening, yielding more reliable and predictive data for preclinical

assessments; however, they necessitate prolonged cell culture periods and present challenges in

accessibility compared to immortalized cell lines.” (line 78)

2.

2. The following literature provides a good explanation of ‘The three-dimensional (3D) models have been shown to overcome these limits by the development of organoids and spheroids’ and may consider being cited : A. Gu, J. Li, S. Qiu, S. Hao, Z.-Y. Yue, S. Zhai, M.-Y. Li, Y. Liu, Pancreatic cancer environment: From patient-derived models to single-cell

omics. Mol. Omics 2024, 20, 220–233. DOI: 10.1039/D3MO00250K

We thank the reviewer for providing this recent and relevant reference. We have cited it and as reference [17] to better explain the significance of using three-dimensional models. Additionally, we have included human xenografts and explants as examples of 3D models alongside organoids and spheroids in the text (highlighted in yellow): ‘

The three-dimensional (3D) models have been shown to overcome these limitations through the development of organoids, spheroids, human xenografts, and explants”.

3.

3. Although the author does not provide application experiments, it is essential to include a discussion on potential application areas of this model and the author's future research

directions at the conclusion of this article.

The potential applications of this model and our future research were briefly mentioned in the initial submission. To provide more detail, we have added a new paragraph in the conclusion (highlighted in yellow in the text): ‘Consequently, the proposed structure not only serves as a valuable tool for investigating the immunomodulatory response within the context of immunotherapeutic agents (whether cellular or material-based) but also as a robust platform for drug screening. This model is capable of assessing the efficacy of both established and novel therapeutic agents within a controlled environment that closely mirrors the complexity of the tumor microenvironment. Moreover, this model can be further enhanced by incorporating immune cells, increasing its biomimicry and allowing for more precise studies of tumor-immune system interactions, which are critical for advancing immunotherapy research. Our research team intends to utilize this model to evaluate the efficacy of gemcitabine, a well-known first-line treatment for PDAC, or a Fofirinox chemotherapy regimen and compare its performance with other in vitro 3D models. Furthermore, the bio-printed model holds potential as a ‘chemogram,’ facilitating personalized medicine by enabling the customization of treatment strategies based on the specific characteristics of an individual’s tumor. The versatility of this model makes it a powerful tool for studying various cancer types, offering critical insights into tumor behavior and treatment responses across different therapeutic modalities.’

We thank you for your consideration of our revised manuscript and look forward to your favorable review. Should you require any further information or clarification, please do not hesitate to contact me.

Reviewer 2 Report

Comments and Suggestions for Authors

In this study, the authors developed a 3D cell culture model including pancreatic cancer cells, stromal cells, and endothelial cells using 3D bioprinting. The authors validated this model could be maintained under specific culture conditions and detected several markers to identify these certain cell culture conditions. Below are some comments and considerations:

The authors explained that a higher concentration of cell density reduced cell numbers,  how did the authors address this issue? Is there any difference in the incubation time and the nutrient composition of the medium that leads to this significance? Besides, the mechanism behind the reduction in cell number, whether it affects proliferation or induces a specific type of cell death, should be explained.

Please define the ratio of cells used in this study. Since MeWo cells grow faster than Panc-1 cells, how did the authors control and define their ratio?

In the aggregates study, the results of cell diameters seems to overlap between the small and large aggregates. How did the authors distinguish them? Is there any statistical significance in Figure 7?

For Figures 3 and 6, please explain what the colors represent, particularly the yellow/brown.

What are these aggregates? Are they aggregated proteins or nonspecific aggregations? Please specify. How these aggregations affect the 3D culture should also be explained.

In Figure 8, the background in the IHC staining from 3D cell culture appears very clean; please confirm, because line 443 describes monoculture, but Figure 8 was 3D culture. The authors should compare the staining of these markers on the first and last days. Furthermore, please clarify if the images were monocultures of each cell type.

In Figure 9, the intensity of the green color (HUVEC) is concentrated on day 1, reduced on day 3, and more diffused on day 7. There is no tube-formation-like structure on day 7. How do the authors support their claim of vascularization?

Overall, do all three cell types compete for growth space? Given that the concentration of the culture medium and oxygen levels vary between the central and superficial regions of the 3D cell culture, how do the authors account for and control PDAC growth from different biological replicates? Additionally, the potential mechanisms leading to the differences in aggregation should be discussed.

Author Response

Reviewer 2

The authors developed a 3D cell culture model including pancreatic cancer cells, stromal cells, and endothelial cells using 3D bioprinting. The authors validated this model could be maintained under specific culture conditions and detected several markers to identify these certain cell culture conditions. Below are some comments and considerations.

  1. The authors explained that a higher concentration of cell density reduced cell numbers, how did the authors address this issue? Is there any difference in the incubation time and the nutrient composition of the medium that leads to this significance? Besides, the mechanism behind the reduction in cell number, whether it affects proliferation or induces a specific type of cell death, should be explained.

We thank the reviewer for raising these important points.

Concerning the incubation time and nutrient composition, we ensured that all experiments were conducted under consistent incubation times and nutrient compositions across different cell densities. Therefore, the reduction in cell numbers observed at higher densities cannot be attributed to variations in these experimental conditions. We acknowledge the importance of understanding the mechanism behind the reduction in cell numbers. According to previous reports a reduction in size over time can be probably related to organizational processes within the spheroids (Dorst et al. 2014). However the mechanism leading to the organization of cells is still not known. We agree that this is an important area for future research. The focus of our study was primarily on the observed outcomes at varying cell densities rather than on dissecting the underlying mechanisms of cell death.

  1. Please define the ratio of cells used in this study. Since MeWo cells grow faster than Panc-1 cells, how did the authors control and define their ratio?

The cell ratios used in this study were predefined as follows: 1:1 for Panc-1/ MeWo and 1:4:4 for Panc-1/MeWo/HUVEC We started with an initial ratio of 1:1 Panc-1/ MeWo, and since MeWo cells proliferate quickly, there will naturally be a higher proportion of CAFs compared to cancer cells in the bio-printed structure over time. This approach does not involve adjusting the ratio throughout the culture period but aims to replicate the tumor microenvironment based on the initially specified ratios.

  1. In the aggregates study, the results of cell diameters seem to overlap between the small and large aggregates. How did the authors distinguish them? Is there any statistical significance in Figure 7?

Small diameter (SD) and large diameter (LD) were obtained using ImageJ image processing software. These are the diameters of the ellipse that pass through the center of the ellipse, where the large axis (Large Diameter, a) and small axis (Small Diameter, b) intersect at the central point of the ellipse.

We appreciate the reviewer’s observation regarding the apparent overlap in cell diameters between the small and large aggregates in the study.

To Distinguish Small and Large Aggregates: Aggregates were classified into “small” and “large” groups based on predefined size criteria established at the beginning of the analysis. This categorization allowed us to separately analyze and compare the growth patterns and responses of these two distinct groups throughout the experiment.

Thereby, to assess the statistical significance of the differences in cell diameters between small and large aggregates, we performed a Mann-Whitney U test at each time point. The calculated p-values are as follows:

  • Day 1: p = 0.0017
  • Day 3: p = 0.0027
  • Day 7: p = 0.0030
  • Day 10: p = 0.0012
  • Day 14: p = 0.0062

These results indicate statistically significant differences at each time point (p < 0.01).

The last sentences highlighted in yellow were added to the legend of the in the Figure7.

  1. For Figures 3 and 6, please explain what the colors represent, particularly the yellow/brown.

For Figures 3 and 6, we performed hematoxylin-eosin-saffron (HES) staining to differentiate various cellular and extracellular components in our samples. Hematoxylin binds to nucleic acids (DNA and RNA) within cell nuclei, resulting in the characteristic purple coloration of the nuclei. This occurs because hematoxylin is a basic dye that interacts with the acidic components of the cell nucleus. Eosin, an acidic dye, binds to the basic (alkaline) proteins and cytoplasmic components, staining the cytoplasm, connective tissue, and extracellular matrix pink. This contrast enhances the visibility of the hematoxylin-stained nuclei against the surrounding cellular structures.

The yellow coloration observed is due to saffron, which specifically stains collagen fibers. This is particularly evident in the hydrogel, which is composed of gelatin, a partially hydrolyzed form of collagen. In the figures, this yellow staining highlights the presence of the hydrogel matrix, clearly distinguishing it from the surrounding tissue components. The brown coloration mostly stands for more pronounced staining of collagen fibers.  These staining techniques allowed us to effectively visualize and differentiate the cellular and extracellular components in our samples, providing clear and detailed images that support our analysis.

In the new version of the MS we added this explanation to the legend to the Figure 6.

  1. What are these aggregates? Are they aggregated proteins or nonspecific aggregations? Please specify. How these aggregations affect the 3D culture should also be explained.

The aggregates observed in our study represent cellular clusters that spontaneously formed over time from initially isolated cells. As shown in Figure 7, this process is clearly evident: on Day 1, the two cell types are distinctly separated, whereas by Day 7, the aggregates are composed of both cell types, as indicated by the HES staining.

The progression from separate cells to integrated aggregates highlights the ability of cells to migrate within the 3D bioprinted structure. By Day 7, the presence of both cell types within the same aggregates demonstrates significant cellular migration and interaction within the hydrogel matrix. These findings suggest that even within the environment of a 3D bioprinted structure, cells can move and self-organize into complex clusters. This migratory behavior is crucial for the formation of the heterogeneous clusters characterized in Figure 7. Finally, the aggregation of cells within the 3D bioprinted structure enhances the physiological relevance of the model by promoting more realistic cell-cell interactions, resistance to stress, and the creation of microenvironments that mimic in vivo conditions.

Concerning the influence of the these aggregations on  the 3D culture:

The formation of cellular aggregates within the 3D bioprinted culture significantly impacts both the structural organization and functional behavior of the cells. These aggregates serve as microenvironments that can mimic in vivo conditions more closely than dispersed cells. First, they can enhance the cell-cell interactions. Indeed, the aggregation of cells within the 3D culture promotes direct cell-cell interactions, which are critical for various physiological processes, including signal transduction, tissue development, and cellular differentiation. These interactions are less pronounced in 2D cultures or when cells are evenly dispersed.

Second, they can increase the cells resistance to environmental stress. The aggregated cells in 3D cultures often exhibit enhanced resistance to environmental stressors, such as nutrient deprivation or hypoxia, compared to isolated cells. This resistance more closely mimics, then, the conditions cells experience within tissues in vivo, where cells rely on neighboring cells for support and survival.

Third, it can recreate the impact on diffusion gradients. Indeed, the formation of aggregates affects the diffusion of nutrients, oxygen, and waste products within the 3D culture. Larger aggregates may develop gradients of these factors, leading to differential cellular behaviors such as varying rates of proliferation or apoptosis depending on the position within the aggregate. This phenomenon is particularly relevant in cancer research, where tumor microenvironments exhibit similar gradients. Finally and more importantly it can help in mimicking the tumor microenvironments which is necessary when considering cancer research. T ability of cells to aggregate within the 3D culture creates a more accurate model of tumor microenvironments. These aggregates can replicate the complex architecture and heterogeneity of tumors, making the 3D culture a valuable tool for studying cancer progression, drug response, and cell migration.

  1. In Figure 8, the background in the IHC staining from 3D cell culture appears very clean; please confirm, because line 443 describes monoculture, but Figure 8 was 3D culture. The authors should compare the staining of these markers on the first and last days. Furthermore, please clarify if the images were monocultures of each cell type.

Thank you for your comment regarding Figure 8. I would like to clarify the following points:

- 3D Culture Setting: Figure 8 indeed presents a 3D culture system, consistent with the bioprinting approach used throughout our study. This figure illustrates the 3D culture in three different conditions: Panc-1 monoculture, MeWo monoculture, and Panc-1/MeWo coculture. This setup allows us to examine how each cell type behaves and interacts within the hydrogel under these varied conditions.

- Marker Assessment on Day 1: We intentionally chose not to perform any staining or labeling on Day 1 of the culture. Our primary objective was to observe whether each cell type maintains its phenotype over time within the hydrogel, and in the case of coculture, how cells interact with each other over time. This approach focuses on the long-term behavior and stability of cell phenotypes rather than their initial states. This longitudinal assessment is crucial for understanding how cells maintain their characteristics and interact in a dynamic three-dimensional environment.

This strategy provides a more comprehensive view of cell behavior over time, which is essential for capturing the full complexity of interactions within the 3D culture. By observing these dynamics without early intervention, we can better understand the natural progression of cell behavior in a 3D context, which is vital for the integrity and relevance of our findings

  1. In Figure 9, the intensity of the green color (HUVEC) is concentrated on day 1, reduced on day 3, and more diffused on day 7. There is no tube-formation-like structure on day 7. How do the authors support their claim of vascularization?

It is indeed premature to discuss the development of a fully functional vascular network at this stage. Our preliminary results suggest that over time, cells begin to organize themselves into network-like structures within the alginate/gelatin hydrogel. However, these cells have not yet progressed to the stage of tubular formation.

In the manuscript, we clearly stated that our efforts toward vascularization were exploratory and that further experimentation is necessary to refine this process. Specifically, while we successfully demonstrated that HUVEC cells can be incorporated into this 3D bioprinting structure and remain viable, the formation of vascularized structures is still in its early stages. The initial sprouting observed indicates potential, but enhancing this process—particularly through the addition of specific growth factors—will be essential for advancing toward fully functional vascular networks.

To accurately assess the progression to a functional vascular network, it will be necessary to extend the culture duration and carefully monitor endothelial cell junctions to ensure their structural integrity. Moreover, verifying whether the cells are transitioning to a fenestrated phenotype, which is crucial for vascular function, will be a key focus of ongoing studies. As you noted, the challenge of identifying and staining these structures becomes increasingly difficult over time due to the progressive loss of fluorescence.

Our preliminary findings are promising but clearly indicate that the development of vascular networks within this 3D bioprinting model requires further refinement and optimization. We highlighted these points in the manuscript to underscore only the feasibility of incorporating HUVEC.

  1. Overall, do all three cell types compete for growth space? Given that the concentration of the culture medium and oxygen levels varies between the central and superficial regions of the 3D cell culture, how do the authors account for and control PDAC growth from different biological replicates? Additionally, the potential mechanisms leading to the differences in aggregation should be discussed.

Thank you for your insightful remarks. All cell types (Panc-1, MeWo and HUVEC) can compete for growth space in the 3D culture system. It has been observed over time that cells move to the periphery of the bioprinted structure due to the oxygen and nutrient gradient and that CAFs organize around cancer cells, thus delimiting each of their growth space. Experimental conditions including seeding density and respect of ratios are always identical as well as homogeneous hydrogel preparation. We always perform a minimum of 3 biological replicates and analyze statistical data to control for variability to ensure minimum of differences between all biological replicates.

We thank you for your consideration of our revised manuscript and look forward to your favorable review. Should you require any further information or clarification, please do not hesitate to contact me.

Sincerely,

Pr. Halima Alem-Marchand

Université de Lorraine

Round 2

Reviewer 2 Report

Comments and Suggestions for Authors

The authors responded and addressed the reviewer's comments. I have no further questions regarding this manuscript.